# The Importance of Immunohistochemistry in the Evaluation of Tumor Depth of Primary Cutaneous Melanoma

**DOI:** 10.3390/diagnostics13061020

**Published:** 2023-03-07

**Authors:** Anca Maria Pop, Monica Monea, Peter Olah, Raluca Moraru, Ovidiu Simion Cotoi

**Affiliations:** 1Faculty of Medicine, George Emil Palade University of Medicine, Pharmacy, Science, and Technology of Târgu Mureș, 540139 Târgu Mureș, Romania; 2Department of Odontology and Oral Pathology, George Emil Palade University of Medicine, Pharmacy, Science, and Technology of Târgu Mureș, 540139 Târgu Mureș, Romania; 3Department of Medical Informatics and Biostatistics, George Emil Palade University of Medicine, Pharmacy, Science, and Technology of Târgu Mureș, 540139 Târgu Mureș, Romania; 4Department of Anatomy and Embryology, George Emil Palade University of Medicine, Pharmacy, Science, and Technology of Târgu Mureș, 540139 Târgu Mureș, Romania; 5Department of Plastic Surgery, County Clinical Hospital Mureș, 540103 Târgu Mureș, Romania; 6Department of Pathophysiology, George Emil Palade University of Medicine, Pharmacy, Science, and Technology of Târgu Mureș, 540139 Târgu Mureș, Romania; 7Department of Pathology, County Clinical Hospital Mureș, 540011 Târgu Mureș, Romania

**Keywords:** primary cutaneous melanoma, Breslow thickness, immunohistochemical markers, tumor staging

## Abstract

Primary cutaneous melanoma (PCM) is the most aggressive skin malignancy, with an increasing incidence and significant mortality. Tumoral invasion, expressed as Breslow thickness, is routinely assessed on hematoxylin and eosin (HE), although this stain may sometimes underestimate the tumoral depth. The aim of this study was to compare the efficiency of the immunohistochemical (IHC) markers S-100, SOX10, Melan-A, and HMB-45 with HE for the evaluation of the Breslow thickness and staging of PCM. This retrospective study included 46 cases of PCM diagnosed between 2015 and 2022; for each case, the Breslow thickness using HE, S-100, SOX10, Melan-A, and HMB-45 was measured and the appropriate T category was recorded. The highest values of the Breslow thickness were observed for S-100. However, S-100, SOX10, and Melan-A provided statistically significant higher values of the Breslow thickness compared to HE, but no difference was noted between HMB-45 and HE. S-100 was most frequently involved in increasing the T category (26.1%), the majority of cases being upstaged from T1a to T1b. The IHC markers S-100, SOX10, and Melan-A contributed to better evaluation of the melanoma invasion, especially in thin melanomas, but their impact on staging and consecutive treatment remains to be confirmed by future studies.

## 1. Introduction

Primary cutaneous melanoma (PCM) originates from melanocytes, which are cells located in the basal layer of the epidermis and in hair follicles and responsible for producing melanin [1]. Despite considerable advances in diagnostic and treatment protocols, melanoma continues to be the major cause of death due to skin malignancies [2,3]. Its incidence is increasing worlwide, as a result of continuous exposure to risk factors (chronic sun damage, aging, history of skin cancer, use of tanning beds), but also due to more accurate diagnostic criteria. However, despite better diagnosis and more efficient treatment, a significant lowering of the melanoma death rate has not yet been observed [4,5]. When identified early, PCM can be cured [5,6]. The majority of tumors arise from melanocytes located at the dermo-epidermal junction, either from normal skin or in association with a nevus [7]. Initially, cells only proliferate inside the epidermis, and in this case the tumor is considered melanoma in situ. After a while, the tumor becomes invasive, extends to the dermis and more profound structures, and then to blood and lymph vessels, therefore aquiring metastatic potential [8]. The risk of metastasis is correlated with the tumoral invasion evaluated as the Breslow thickness, but also with other factors such as ulceration and mitotic rate [9,10]. Theoretically, melanoma in situ does not metastasize, but there have been cases reported of metastatic disease, probabably due to unidentified small clusters of tumoral cells in the dermis or as a consequence of regression [11].

Despite interobserver discrepancies, the Breslow thickness is the most reproducible histopathological parameter, with vital importance for melanoma staging and prognosis, and is used to define surgical excision margins and indicate the necessity for sentinel lymph node biopsy [12,13,14,15]. Its prognostic role is emphasized in the 8th edition of the American Joint Committee on Cancer (AJCC) staging manual, in which the Breslow thickness categorized as a dichotomized variable (<0.8 mm and 0.8–1.0 mm) proved to be a more reliable prognostic factor compared to the mitotic rate [16]. The risk of developing metastases has also been evaluated based on more sophisticated pathological features, such as lymphovascular invasion, tumor infiltrating lymphocytes (TILs), or mitotic rate, which showed inconsistent results among experts, due to differences in training or experience [17]. The mitotic rate, previously used for characterizing the T1 category, was discordantly reported between centers; furthermore, the Clark level of invasion showed a wider variability between pathologists because it is a categorical variable [17,18]. 

Hematoxylin and eosin (HE) is the standard stain for the evaluation of the Breslow thickness, but it can sometimes lead to overlooking isolated tumoral cells located in the dermis, or it can underestimate the tumoral invasion in cases associated with regression [19,20,21]. A few studies have reported the use of immunohistochemical (IHC) techniques in melanoma thickness assessment [22,23]; however, their routine application in the staging of PCM is not well established in the scientific literature. The aim of this study was to compare the efficiency of four IHC markers specific for melanocytic differentiation: S-100, SOX10, Melan-A, and HMB-45, with the gold standard stain HE in the evaluation of the Breslow thickness and staging of PCM. The null hypothesis to be tested was that there are no statistically significant differences between the measurement of the Breslow thickness with the previously mentioned IHC markers and HE. 

## 2. Materials and Methods

### 2.1. Study Design and Case Selection 

This retrospective study was conducted in the Department of Pathology of the County Clinical Hospital Mureș between January 2021 and June 2022 and included all cases of PCM diagnosed in this department beginning in 2015. The criteria for inclusion in the study were invasive melanoma with complete IHC profile (S-100, SOX 10, HMB-45, Melan-A). The following exclusion criteria were applied: metastatic or desmoplastic melanoma, inadequate depth of the biopsy specimen, local re-excisions, PCMs associated with a nevus, and uninterpretable IHC reactions. The study was conducted in accordance with the Declaration of Helsinki and was approved by the Ethics Committee of the hospital (protocol number 16381/06.01.2021). For each case, five slides with 3 µm thickness were selected, stained with HE, S-100 (polyclonal antibody, Ventana, AZ, USA), SOX10 (SP267, primary monoclonal antibody, Cell Marque, CA, USA), HMB-45 (monoclonal antibody, Ventana, AZ, USA), and Melan-A (monoclonal antibody, Ventana, AZ, USA). Furthermore, all sections had previously been obtained from the same paraffin block. The process of case selection is illustrated in Figure 1.

### 2.2. Histopathologic Evaluation of the Specimens 

The specimens were coded, randomized, and then reevaluated with an optical microscope (Zeiss Primo Star, Carl Zeiss Meditec AG, Jena, Germany) in a single-blind manner by an experienced senior dermatopathologist (O.S.C.), who had been evaluated for intraobserver reliability (intraclass correlation coefficient 0.87, showing very good agreement). The Breslow thickness and the Clark level of invasion were recorded for each case. The Breslow thickness was measured from the top of the granular layer or the base of the ulceration to the deepest situated tumoral cell. PCMs located exclusively in the dermis were measured in a standard manner (from the top of the granular layer of the epidermis). The Breslow thickness was evaluated with the ×5 objective, and the isolated tumoral cells located in the dermis were identified with the ×20 objective. The Breslow thickness was measured using ZEN Blue software (Carl Zeiss Microscopy GmbH, Jena, Germany), with a precision of 0.1 mm, as recommended by the 8th edition of the AJCC staging system. Tumors with a thickness ≤1 mm were evaluated with 0.01 mm precision but the result was rounded to a single decimal. After the measurement of the Breslow thickness on the 5 histological sections, the appropriate T category was recorded for each case. 

### 2.3. Statistical Analysis

The patients’ demographic data were assessed using descriptive statistics (Microsoft Excel, Microsoft, WA, USA). The histopathological parameters were statistically analyzed using GraphPad Prism 7 software (GraphPad Software, San Diego, CA, USA). Continuous variables were presented as mean ± standard deviation for normally distributed data and as medians for non-normally distributed data, respectively. Categorical variables were reported as numbers and percentages. Data were evaluated for normal distribution based on the Kolmogorov-Smirnov test and then were statistically analyzed using the Friedman test. A post hoc analysis was performed using Dunn’s multiple comparisons test. The kappa coefficient was used for evaluation of the agreement rate between the applied staining techniques in the staging of PCM. The level of statistical significance was set at a value of *p* < 0.05 (two-tailed) with a 95% confidence interval. 

## 3. Results

This retrospective study included 46 cases of PCM diagnosed between 2015 and 2022, in 25 men (54.3%) and 21 women (45.7%) with a mean age of 61.7 ± 17.4 years (range 25–84). Important histopathological data are summarized in Table 1. 

The Breslow thickness values categorized according to the 8th edition of the AJCC criteria are presented in Table 2.

A detailed description of the 46 cases included in the study is presented in Table 3. Based on the Friedman test, there was a statistically significant difference between the values of the Breslow thickness measured on HE and on the four IHC markers (*p* < 0.0001). The lowest Breslow thickness values were recorded for HE and HMB-45 and the highest for S-100 (Table 3).

Based on the post hoc analysis, it was proven that the values of the Breslow thickness obtained with S-100, SOX10, and Melan-A were significantly higher than those obtained with HE, with the greatest difference being observed for S-100 and SOX10 (*p* < 0.001). Moreover, the values obtained with S-100 were also significantly higher than those obtained with HMB-45 (*p* < 0.001) and Melan-A (*p* < 0.05). The values recorded with HMB-45 did not significantly differ from those recorded with HE (*p* > 0.05). There was no significant difference between SOX10 and the markers S-100 and Melan-A (*p* > 0.05) (Table 4).

The variation of the Breslow thickness obtained with HE and the IHC markers is illustrated in Figure 2. The majority of the cases in which the tumoral invasion was deeper with IHC staining compared to HE (at least 0.1 mm) was recorded for S-100 (80.4%). In 47.8% of the cases HE and HMB-45 identified an identical depth of tumoral invasion. The combined use of the four IHC markers led to a more profound tumoral invasion compared to HE in 29 lesions (63.1%).

The agreement rates between the T categories obtained based on HE staining and IHC techniques are summarized in Table 5. 

The kappa correlation coefficients were interpreted as follows: substantial for S-100 (0.601) and SOX10 (0.627) and very good for HMB-45 (0.896) and Melan-A (0.813). Upstaging was noted in 12 cases (26.1%) after applying S-100, in 9 cases (19.6%) after applying SOX10, in 2 cases (4.3%) after applying HMB-45, and in 3 cases (6.5%) after applying Melan-A.

## 4. Discussion

Our results showed that the Breslow thicknesses obtained with the IHC markers S-100, SOX10, and Melan-A had statistically significant higher values than those obtained with HE. Moreover, the combined use of the four studied IHC markers found a deeper tumoral invasion in 63% of the lesions, therefore rejecting the null hypothesis. Regarding the T category, the most frequent upstaging was based on S-100 and mostly occurred from T1a to T1b. This may have major implications in clinical practice, as patients with T1b melanoma can be candidates for sentinel lymph node biopsy, with an important impact on their diagnosis and treatment [24,25]. The Breslow thickness is considered the “bedrock” for melanoma staging, and its precise measurement requires a complete excision [17], as other less-invasive techniques used for the diagnosis of malignant lesions, such as shave biopsy or exfoliative cytology did not provide reliable results [26,27]. Previous studies reported a worse prognosis in patients with tumoral thickness >0.75 mm [28,29,30], which led to the establishment of the 0.8 mm thickness as a cut-off value in the 8th edition of the AJCC staging manual [31]. Although thin melanomas (considered ≤1 mm) usually have a generally accepted good prognosis, the survival rate decreases with every 0.1 mm increase in the tumor thickness [32]. Conversely, for ultrathin melanomas (≤0.5 mm) the survival rate at 10-year-follow-up was over 99%, and the diagnosis in this early stage enabled a marked decrease of the risk for distant metastasis [33]. Richetta et al. [34] confirmed the essential role of the Breslow thickness as the most powerful factor in the prediction of metastatic disease, as in their study all thin melanomas that developed lymph node metastases exhibited a Breslow thickness >0.6 mm. Furthermore, accurate identification of patients with thin melanomas ranging around the value of 0.8 mm is also particularly important for appropriate monitoring of these patients, as late, rather than early, mortality is more frequently observed [35]. 

In a study published in 2019, Bishop and Tallon [14] compared the precision of measuring the Breslow thickness in one, three, five, and 10 levels, the authors observing an important benefit when examining three instead of one level. Besides the challenges regarding the measurement of T1 melanomas, another problematic category is represented by melanomas in situ, which can be considered invasive after examining multiple slides. Due to the previously mentioned reasons, the use of IHC markers has markedly increased in recent years, especially in lesions with atypical histopathological features and in the evaluation of thin melanomas [36]. Immunohistochemistry is extensively requested by dermatopathologists, not only for reassuring the melanocytic origin in case of poorly differentiated lesions, but also for confirming surgical excision margins and for the assesment of the tumoral depth when concomitant regression or fibrosis are present [37]. Other reported pitfalls in the accurate diagnosis of thin melanomas exclusively on HE slides are the presence of isolated nests of tumoral melanocytes in the dermis, macrophages, dense lichenoid inflammation, or marked lymphocytic infiltrate, which can blur the dermo-epidermal junction [19]. 

Among the many proposed IHC markers used in the diagnosis of PCM, S-100, SOX10, Melan-A, and HMB-45 are frequently applied [3,38]. The expression of S-100 in PCM is well-known; however, the lower specificity of this marker requires its use in combination with HMB-45 or Melan-A [3,39]. Moreover, it may be less appropriate for lesions confined to the epidermis, as S-100 negative cases of invasive melanoma have also been described in the scientific literature [40]. In our study, the highest values of Breslow thickness were recorded for S-100, which may have been due to the fact that S-100 is regarded as a very sensitive IHC marker (sensitivity ranging between 93–100%) [2,3,19,41]. SOX10 is a nuclear transcription factor with vital importance in the differentiation of progenitor cells of the neural crest into melanocytes [42,43]. Its nuclear staining pattern makes it more suitable for the evaluation of melanoma, as it allows differentiating between immunodye and melanin located in melanophages and keratinocytes [19]. However, it can also stain benign melanocytes; therefore, the results must be cautiously interpreted in lesions associated with a nevus [44]. HMB-45 is a monoclonal antibody directed against PMEL17 (also referred to as gp100), with lower sensitivity, but better specificity for melanocytic differentiation [2,45]. It is useful in distinguishing between an invasive melanoma and a nevus, as it only stains the superficial nevic cells, due to loss of expression associated with maturation, and is regarded as the most appropriate marker for the evaluation of the junctional component [45,46]. Although false positive results of HMB-45 have previously been reported, Ordóñez [47] found that HMB-45 is negative in tumors with glial, mesenchymal, lymphoid, or epithelial origin. Melan-A/MART 1 is expressed in the endoplasmic reticulum and melanosomes, being considered a more sensitive marker than HMB-45. Although the staining of intensely pigmented keratinocytes is possible with Melan-A, but not with SOX10, Dass et al. [45] did not find a significant difference between the staining with these two markers, this observation also being in accordance with the results of our study (*p* > 0.05 for the values of the Breslow thickness measured based on SOX10 and Melan-A). Melan-A can aid in the identification of isolated tumoral melanocytes located in the dermis, which can upstage a previously diagnosed melanoma in situ with HE to an invasive lesion. Drabeni et al. [23] found a higher Breslow thickness in nearly 60% of cases when applying Melan-A in comparison to HE. Megahed et al. [48] reported the presence of dermal invasion in 30 cases out of 104 melanomas considered in situ lesions on HE. None of the 74 cases identified as in situ lesions using both staining techniques developed distant metastasis, but two cases identified as invasive only using Melan-A were associated with metastatic disease. However, the occuerence of pseudomelanocytic nests, known as aggregates of Melan-A positive cells at the dermoepidermal junction, may represent a diagnostic challenge in melanoma in situ associated with lichenoid inflammation. Clinical data, which are crucial in these cases, are not always available to pathologists; therefore, the concomitant use of nuclear stainings such as MITF (microphthalmia-associated transcription factor) and SOX10 has been proposed [49]. SOX10 exhibited higher specificity (96%) compared to Melan-A (17%) in evaluating epidermal malnocytes and consequently in avoiding overdiagnosis of melanoma in situ in sun-damaged skin [50]. Furthermore, the concomitant use of Melan-A with markers such as HMB-45 or S-100 was also recommended in order to avoid the misdiagnosing of “pseudomelanocytic” cells as melanoma cells [38,45]. 

Based on our results, we identified statistically significant higher values of Breslow thickness using the more sensitive IHC markers S-100 (*p* < 0.001), SOX 10 (*p* < 0.01), and Melan-A (*p* < 0.05), which is in accordance with the findings of Kamyab-Hesary et al. [22]. Recently, other markers have been also been proposed for assesing the Breslow thickness in challenging cases. For example, PRAME (preferentially expressed antigen in melanoma) proved to be useful in evaluating melanoma in situ associated with melanocytic cells exhibiting nevoid features or in cases of invasive tumors associated with dermal nevic cells. Moreover, PRAME usually stains melanoma cells and is negative in non-tumoral melanocytes, in comparison with Melan-A and SOX10 that stain all melanocytes [51]. MITF was also identified as an adjuvant diagnostic tool, especially in lesions arrising in sun-damaged skin, in which Melan-A can lead to overdiagnosis [52].

However, despite their possible contribution to the improvement of the tumoral depth evaluation, the role of the IHC markers in the staging and, consecutively, in the treatment planing of PCM remains a matter of debate, as all currently available treatment strategies and prognostic models rely on histopathologic assessment of the tumoral invasion based on the standard staining with HE [53].

### Strenghts, Limitations, and Future Research Directions 

The importance of our study rests in the simultaneous evaluation of the Breslow thicknesss using four widely applied immunohistochemical markers for melanoma diagnosis, compared to other studies that included fewer immunohistochemical markers. Furthermore, almost half of the the selected cases were thin melanomas, in which the precise evaluation of the tumor inavasion is difficult. However, the statistical power of our study is limited by the small sample, originating from a single tertiary center. Moreover, the clinical relevance of the study could have been enhanced by a correlation with the clinical evolution of the patients regarding the occurrence of nodal or distant metastasis. 

## 5. Conclusions

The evaluation of the Breslow thickness exclusively using HE may underestimate the real depth of the tumoral invasion, especially in thin melanomas. Out of the IHC markers included in our research, S-100, SOX10, and Melan-A contributed to a more precise assessment of the tumoral thickness; however, their impact in clinical practice remains to be confirmed by large-scale, multicentric studies. 

## Figures and Tables

**Figure 1 diagnostics-13-01020-f001:**
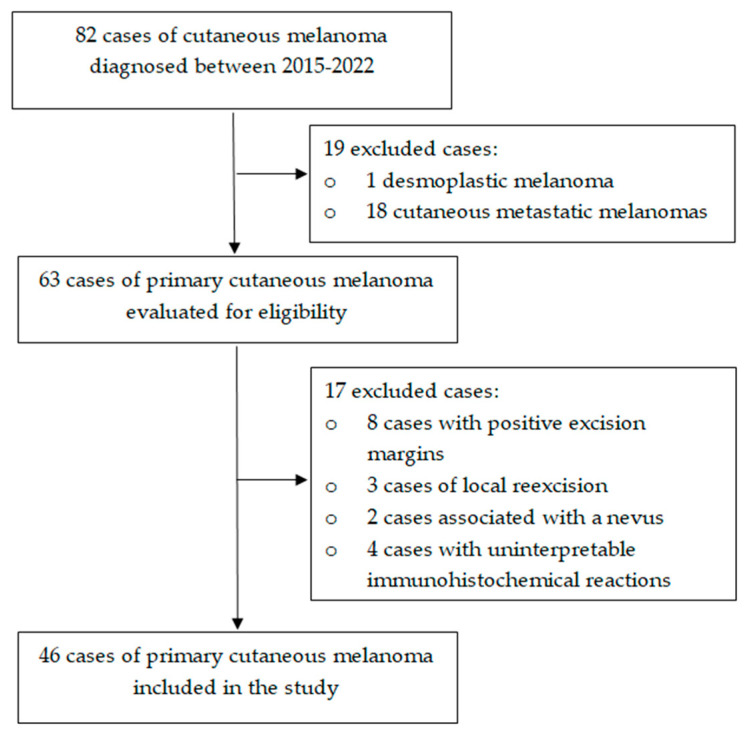
Flow chart illustrating the case selection for the study.

**Figure 2 diagnostics-13-01020-f002:**
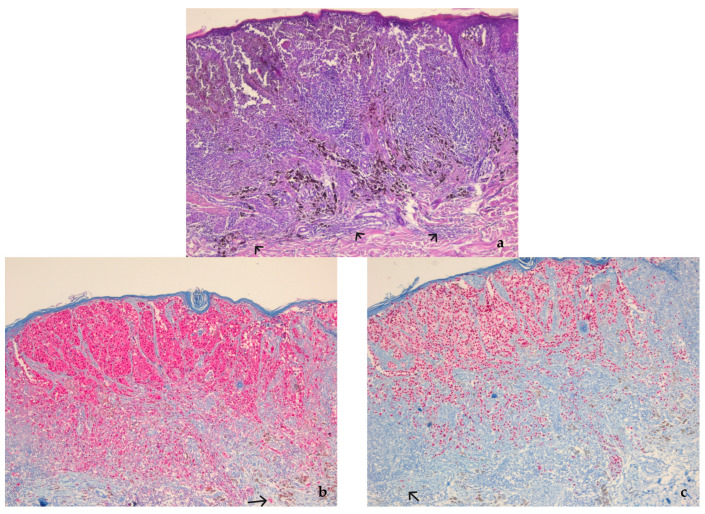
(**a**) Nodular melanoma, stage pT1a, in a 65-year-old male patient. The tumor is in the vertical growth phase; pagetoid spread and non-brisk TILs can be observed. The value of the Breslow thickness measured on HE was <0.8 mm; Clark level of invasion II. (HE, ×5). (**b**) The value of the Breslow thickness measured on S-100 was 1.0 mm, upstaging the tumor to pT1b. (S-100, ×5); (**c**) The value of the Breslow thickness measured on SOX10 was 1.0 mm. (SOX10, ×5). (**d**) The value of the Breslow thickness measured on HMB-45 was 0.9 mm. (HMB-45, ×5). (**e**) The value of the Breslow thickness measured on Melan-A was 0.9 mm. (Melan-A, ×5). Isolated tumor cells suggesting invasion are highlighted with black arrows (case from the archive of the County Clinical Hospital Mureș).

**Table 1 diagnostics-13-01020-t001:** Histopathological features of the evaluated cases.

Variables		Frequency
Histological subtype	In situ	5 (10.9%)
	Nodular	26 (56.5%)
	Superficial spreading	10 (21.8%)
	Lentigo maligna melanoma	2 (4.3%)
	Acral lentiginous	3 (6.5%)
Ulceration	Present	14 (30.4%)
	Absent	27 (58.7%)
	Not applicable	5 (10.9%)
Clark level on invasion	I	5 (10.9%)
	II	11 (23.9%)
	III	8 (17.4%)
	IV	17 (36.9%)
	V	5 (10.9%)

**Table 2 diagnostics-13-01020-t002:** The values of the Breslow thickness recorded on HE and IHC markers summarized according to the AJCC criteria.

Breslow Thickness	Staining				
	HE	S-100	SOX10	HMB-45	Melan-A
In situ	5	3	3	4	3
<0.8 mm	10	9	11	11	12
0.8–1 mm	5	7	5	5	4
>1–2 mm	7	6	8	7	8
>2–4 mm	8	9	8	8	8
>4 mm	11	12	11	11	11

**Table 3 diagnostics-13-01020-t003:** Detailed clinicopathological characteristics of the evaluated cases.

Case	Localization	Histological Subtype	Breslow Thickness (mm)					Clark Level	Mitotic Rate (/mm^2^)	Ulceration
			HE	S-100	SOX10	HMB-45	Melan-A			
1	Trunk	In situ (superficial)	0	0	0	0	0	I	0	N/A
2	Head and neck	In situ (superficial)	0	0	0	0	0	I	0	N/A
3	Head and neck	In situ (superficial)	0	0	0	0	0	I	0	N/A
4	Trunk	In situ (superficial)	0	0.2	0.1	0	0.1	I	0	N/A
5	Trunk	In situ (superficial)	0	0.2	0.1	0.1	0.1	I	1	N/A
6	Limbs	Superficial	0.5	0.7	0.6	0.6	0.7	II	1	Absent
7	Trunk	Superficial	0.5	0.7	0.7	0.5	0.6	II	1	Absent
8	Head and neck	Superficial	0.5	0.7	0.7	0.5	0.6	II	1	Absent
9	Trunk	Lentigo maligna	0.6	0.7	0.7	0.7	0.7	II	1	Absent
10	Limbs	Nodular	0.6	0.7	0.8	0.6	0.7	III	2	Absent
11	Head and neck	Superficial	0.6	0.7	0.7	0.7	0.7	II	1	Absent
12	Trunk	Superficial	0.6	0.8	0.6	0.6	0.7	III	1	Absent
13	Limbs	Nodular	0.7	0.8	0.7	0.7	0.8	III	2	Absent
14	Trunk	Superficial	0.6	0.9	0.8	0.7	0.7	II	1	Absent
15	Trunk	Nodular	0.7	1	1	0.9	0.9	II	2	Absent
16	Head and neck	Lentigo maligna	0.8	1.1	0.7	0.9	0.7	II	1	Absent
17	Head and neck	Superficial	0.8	0.7	0.7	0.7	0.7	II	3	Absent
18	Trunk	Nodular	0.9	1.1	1.1	0.9	1.1	III	2	Absent
19	Head and neck	Nodular	0.9	0.9	0.9	0.8	0.9	II	1	Absent
20	Limbs	Nodular	1	1	1.1	0.9	1	II	2	Absent
21	Trunk	Superficial	1.1	1	1	1.1	1.1	III	2	Absent
22	Trunk	Nodular	1.4	1.5	1.6	1.6	1.5	III	2	Absent
23	Limbs	Nodular	1.6	1.9	1.8	1.7	1.8	III	4	Absent
24	Head and neck	Nodular	1.7	2	2	1.7	1.8	IV	3	Absent
25	Limbs	Nodular	1.8	2.1	1.9	1.8	1.9	IV	7	Present
26	Extremities	Nodular	1.8	2.1	1.9	1.9	1.9	III	6	Absent
27	Trunk	Superficial	1.9	2.2	2.1	1.9	2	IV	3	Absent
28	Trunk	Nodular	2.1	2	2	2.1	2.1	IV	5	Present
29	Trunk	Nodular	2.4	2.5	2.5	2.3	2.4	IV	14	Present
30	Limbs	Nodular	2.7	2.9	2.8	2.8	2.8	IV	7	Absent
31	Limbs	Nodular	2.9	3.1	3.1	2.9	3	IV	8	Absent
32	Extremities	Nodular	3.4	3.7	3.6	3.3	3.5	IV	4	Present
33	Extremities	Nodular	3.5	3.7	3.8	3.6	3.7	IV	6	Absent
34	Limbs	Nodular	3.7	3.8	3.9	3.6	3.8	IV	4	Present
35	Trunk	Nodular	3.8	4.2	4.2	3.9	4	IV	2	Absent
36	Trunk	Nodular	4.1	4.2	4	4.1	4.1	IV	7	Present
37	Trunk	Superficial	4.7	4.8	4.8	4.7	4.8	V	7	Present
38	Limbs	Nodular	6.2	6.5	6.4	6.4	6.2	IV	5	Present
39	Trunk	Nodular	6.7	7.3	7.1	6.7	6.9	IV	11	Present
40	Extremities	Acral	7.4	7.6	7.5	7.3	7.5	V	4	Present
41	Trunk	Nodular	7.4	7.2	7.4	7.3	7.6	IV	6	Absent
42	Extremities	Acral	7.8	8.1	7.9	7.9	7.8	V	47	Present
43	Extremities	Acral	8.2	8.6	8.4	8.2	8.3	V	12	Present
44	Limbs	Nodular	8.4	8.8	8.6	8.4	8.5	IV	7	Absent
45	Trunk	Nodular	9.8	10.1	9.9	9.8	9.9	IV	9	Present
46	Head and neck	Nodular	12.6	12.8	12.7	12.5	12.6	V	11	Present
Median of the Breslow thickness			1.65	1.95	1.85	1.7	1.8			
*p* value			<0.0001 *							

* statistically significant difference based on a nonparametric Friedman test; N/A = not applicable.

**Table 4 diagnostics-13-01020-t004:** Results of the post hoc analysis based on Dunn’s multiple comparison test.

Comparison	*p* Value
HE versus S-100	<0.001 *
HE versus SOX10	<0.001 *
HE versus HMB-45	>0.05
HE versus Melan-A	<0.01 *
S-100 versus SOX10	>0.05
S-100 versus HMB-45	<0.001 *
S-100 versus Melan-A	<0.05 *
SOX10 versus HMB-45	<0.001 *
SOX10 versus Melan-A	>0.05
HMB-45 versus Melan-A	<0.05 *

* statistically significant difference.

**Table 5 diagnostics-13-01020-t005:** Agreement rate between the T category obtained on HE and IHC markers.

		**S-100**						**Kappa Coefficient** **(95% Confidence Interval)**
		**pTis**	**pT1a**	**pT1b**	**pT2**	**pT3**	**pT4**	
**HE**	**pTis**	3	2	0	0	0	0	0.601 (0.44–0.762)
	**pT1a**	0	6	4	0	0	0	
	**pT1b**	0	1	2	2	0	0	
	**pT2**	0	0	1	3	3	0	
	**pT3**	0	0	0	1	6	1	
	**pT4**	0	0	0	0	0	11	
		**SOX10**						
		**pTis**	**pT1a**	**pT1b**	**pT2**	**pT3**	**pT4**	
**HE**	**pTis**	3	2	0	0	0	0	0.627 (0.467–0.786)
	**pT1a**	0	7	3	0	0	0	
	**pT1b**	0	2	1	2	0	0	
	**pT2**	0	0	1	5	1	0	
	**pT3**	0	0	0	1	6	1	
	**pT4**	0	0	0	0	1	10	
		**HMB-45**						
		**pTis**	**pT1a**	**pT1b**	**pT2**	**pT3**	**pT4**	
**HE**	**pTis**	4	1	0	0	0	0	0.896 (0.799–0.993)
	**pT1a**	0	9	1	0	0	0	
	**pT1b**	0	1	4	0	0	0	
	**pT2**	0	0	0	7	0	0	
	**pT3**	0	0	0	0	8	0	
	**pT4**	0	0	0	0	0	11	
		**Melan-A**						
		**pTis**	**pT1a**	**pT1b**	**pT2**	**pT3**	**pT4**	
**HE**	**pTis**	3	2	0	0	0	0	0.813 (0.687–0.939)
	**pT1a**	0	8	2	0	0	0	
	**pT1b**	0	2	2	1	0	0	
	**pT2**	0	0	0	7	0	0	
	**pT3**	0	0	0	0	8	0	
	**pT4**	0	0	0	0	0	11	

## Data Availability

The data presented in this study are available on reasonable request from the corresponding author.

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
