# Peer review of "The Importance of Immunohistochemistry in the Evaluation of Tumor Depth of Primary Cutaneous Melanoma"

_diagnostics, 2023, doi:10.3390/diagnostics13061020_

Round 1

Reviewer 1 Report

I believe that highlighting the invasion on the microscopic images is relevant for those who will read the article. I recommend this change.

The article is very interesting both by discussing the Breslow score, included in the TNM staging (8th) of the patient with malignant melanoma of the skin by the classic HE evaluation compared to the evaluation using immunohistochemistry techniques, but especially by the data obtained in the evaluation of the Breslow score by certain markers (S100, HMB45, SoX10, Melan_A). The purpose of the authors is to evaluate the accuracy of the Breslow score obtained by the two methods (HE and IHC). Although there are several studies in this direction, this article evaluates several possible markers that can be used in establishing the proposed goal and presents data with statistical value for S-100, SOX10 and Melan-A contributed to the better evaluation of the melanoma invasion. The method is very clearly described presenting the inclusion and exclusion criteria as well as the techniques used. For a better understanding of the evaluation of the Breslow score, I think that highlighting the invasion on the microscopic images would be useful. The results and conclusions obtained are very important from the perspective of the underestimation of malignant melanoma invasion in HE microscopic analysis. The change in the staging of oncological patients is associated with the change in their treatment. In certain cases the protocol is just wait and watch and the immunohistochemical evaluation may increase the value of the Breslow score and the patient should benefit from oncological treatment. The conclusions are relevant and meet the hypothesis of the study.

Author Response

Dear Reviewer,

Thank you for evaluating our paper, in the attachment you may find our response to your comments. 

Reviewer 2 Report

Diagnostics (ISSN 2075-4418)

The following is an overview of the article A method for measuring the true temperature of inner surface with cavity effect (diagnostics-2228362). In this study, author(s) proposes the aim of this study was to compare the efficiency of the immuno-21 histochemical (IHC) markers S-100, SOX10, Melan-A and HMB-45 with HE in the evaluation of the 22 Breslow thickness and staging of PCM. The manuscript has contributions to the area of tumor staging and primary cutaneous melanoma.

The author(s) stated in the first part of the study; Primary cutaneous melanoma (PCM) originates from melanocytes, which are cells located in the basal layer of the epidermis and in hair follicles, responsible for producing melanin. Despite the considerable advances in the diagnostic and treatment protocols,  melanoma continues to be the major cause of death due to skin malignancies. Its 4incidence is still increasing worlwide, as a result of continuous exposure to risk factors  (chronic sun damage, aging, history of skin cancer, use of tanning beds), but also due to  more accurate diagnostic criteria. However, despite better diagnosis and more efficient treatment, a significant lowering of the melanoma death rate has not been observed yet. When identified as early as possible, PCM can be cured. The majority of the tumors arise from melanocytes located at the dermo-epidermal junction, either from normal skin or in association with a nevus. Initially, cells proliferate only inside the epidermis and in this case the tumor is considered melanoma in situ. After a while, the tumor becomes invasive, extends to the dermis and more profound structures and then 49 to blood and lymph vessels, therefore aquiring a metastatic potential. The risk of  metastasis is correlated with the tumoral invasion evaluated as the Breslow thickness, but also with other factors such as ulceration and mitotic rate. Theoretically, melanoma in situ does not metastasize, but there have been reported cases of metastatic disease, probabably due to unidentified small clusters of tumoral cells in the dermis or as a consequence of regression. Primary cutaneous melanoma (PCM) is the most aggressive skin malignancy, with increasing incidence and significant mortality. The tumoral invasion expressed as Breslow thickness is routinely assessed on hematoxylin and eosin (HE), although this stain may sometimes underes timate the tumoral depth. The aim of this study was to compare the efficiency of the immuno histochemical (IHC) markers S-100, SOX10, Melan-A and HMB-45 with HE in the evaluation of the Breslow thickness and staging of PCM. This retrospective study included 46 cases of PCM diagnosed between 2012-2022; for each case the Breslow thickness on HE, S-100, SOX10, Melan-A and HMB-45 was measured and the appropriate T category was recorded. The highest values of the Breslow thickness were observed for S-100. However, S-100, SOX10 and Melan-A provided statistically significant higher values of the Breslow thickness compared to HE, but no difference was noted between HMB-45 and HE. S-100 was most frequently involved in increasing the T category (26.1%), the majority of cases being upstaged from T1a to T1b. The IHC markers S-100, SOX10 and Melan-A contributed to the better evaluation of the melanoma invasion, especially in thin melanomas, but their impact on the staging and consecutive treatment remains to be confirmed by further studies.

The author(s) stated in the last part of the study; the evaluation of the Breslow thickness exclusively on HE may underestimate the  real depth of the tumoral invasion, especially in thin melanomas. Out of the IHC markers included in our research, S-100, SOX10 and Melan-A contributed to a more precise assessment of the tumoral thickness, however their impact in clinical practice remains to be confirmed by large-scale, multicentric studies..

However, some points must be highlighted so that the author(s) can review and submit in another round of review: The following corrections are considered to be beneficial for the strengthening of the article.

1. The Conclusions should be reviewed again. The original aspect of the study and its difference from other studies should be clearly explained. (The conclusion should be explored better and it needs to contemplate the eventual restrictions of the developed technique to address future works in this area.)

2. The abstract must be make strong. Abstract should be reviewed again.

3. Some sentences have spelling errors. (Punctuation marks, spaces, etc.). Some places should be left space.

4. It has been a comprehensive study in the literature in recent years. If there are more current literature studies, these should be examined in detail and added to the literature section (Especially, primary cutaneous melanoma; Breslow thickness; immunohistochemical markers;  tumor staging studies.).

5. The authors should compare the results of their method with those of previous studies. As mentioned in the literature, there are several methods with very high accuracy, even better than the proposed method. Author(s) can do compare table (A new table can add about previous studies to result section.). This subject is very important.

6. The motivations of the proposed method are not clear. Which problem does the proposed method attempt to solve? Why the other existing diagnosis methods failed to solve it? What are the advantages of the proposed method compared to other methods? Those should be illustrated more clearly.

7. Carefully check all grammatical error. Still, the English language should be improved. I suggest asking for help from a native English

I think it is ACCEPTABLE after the MAJOR Revisions mentioned.

Author Response

(The authors gave the same response as above.)

Reviewer 3 Report

This is true that role of immunohistochemistry is very important in exact characterization of the type of malignancy. Using S-100, SOX10 and Melan-A for the precise assessment of the tumoral thickness seems to be interesting.  

Author Response

(The authors gave the same response as above.)

Round 2

Reviewer 2 Report

Diagnostics | An Open Access Journal from MDPI

Dear Editor;

The author(s) made all the corrections mentioned (Diagnostics-2228362 - The importance of immunohistochemistry in the evaluation of tumoral depth of primary cutaneous melanoma).

The length of the paper is enough in terms of a scientific paper. Considering studies conducted and results obtained, it is believed that the paper is eligible to be published in your journal after your approval.

I think it is ACCEPTABLE  in your journal after your approval as editor.
